# Deep Sequencing of Small RNAs from Neurosurgical Extracellular Vesicles Substantiates miR-486-3p as a Circulating Biomarker that Distinguishes Glioblastoma from Lower-Grade Astrocytoma Patients

**DOI:** 10.3390/ijms21144954

**Published:** 2020-07-13

**Authors:** Susannah Hallal, Saeideh Ebrahim Khani, Heng Wei, Maggie Yuk Ting Lee, Hao-Wen Sim, Joanne Sy, Brindha Shivalingam, Michael E. Buckland, Kimberley L. Alexander-Kaufman

**Affiliations:** 1Department of Neurosurgical Services, Chris O’Brien Lifehouse, Camperdown 2050, Australia; susannah.hallal@lh.org.au (S.H.); brindha.shivalingam@lh.org.au (B.S.); 2Discipline of Pathology, School of Medical Sciences, The University of Sydney, Camperdown 2006, Australia; Saeideh.EbrahimKhani@health.nsw.gov.au (S.E.K.); michael.buckland@sydney.edu.au (M.E.B.); 3Brainstorm Brain Cancer Research, Brain and Mind Centre, The University of Sydney, Camperdown 2050, Australia; heng.wei@sydney.edu.au (H.W.); maggie.lee@sydney.edu.au (M.Y.T.L.); 4Neuropathology Department, Royal Prince Alfred Hospital, Camperdown 2050, Australia; joanne.sy@health.nsw.gov.au; 5Department of Medical Oncology and NHMRC Clinical Trials Centre, Chris O’Brien Lifehouse, Camperdown 2050, Australia; haowen.sim@lh.org.au; 6Central Clinical School, The University of Sydney, Camperdown 2006, Australia; 7The Kinghorn Cancer Centre, St Vincent’s Hospital, Darlinghurst 2010, Australia

**Keywords:** glioblastoma, extracellular vesicle, glioma, small RNA, next generation sequencing, miRNA, piRNA

## Abstract

Extracellular vesicles (EVs) play key roles in glioblastoma (GBM; astrocytoma grade IV) biology and are novel sources of biomarkers. EVs released from GBM tumors can cross the blood-brain-barrier into the periphery carrying GBM molecules, including small non-coding RNA (sncRNA). Biomarkers cargoed in circulating EVs have shown great promise for assessing the molecular state of brain tumors in situ. Neurosurgical aspirate fluids captured during tumor resections are a rich source of GBM-EVs isolated directly from tumor microenvironments. Using density gradient ultracentrifugation, EVs were purified from cavitron ultrasonic surgical aspirate (CUSA) washings from GBM (*n* = 12) and astrocytoma II-III (GII-III, *n* = 5) surgeries. The sncRNA contents of surgically captured EVs were profiled using the Illumina^®^ NextSeq^TM^ 500 NGS System. Differential expression analysis identified 27 miRNA and 10 piRNA species in GBM relative to GII-III CUSA-EVs. Resolved CUSA-EV sncRNAs could discriminate serum-EV sncRNA profiles from GBM and GII-III patients and healthy controls and 14 miRNAs (including miR-486-3p and miR-106b-3p) and cancer-associated piRNAs (piR_016658, _016659, _020829 and _204090) were also significantly expressed in serum-EVs. Circulating EV markers that correlate with histological, neuroradiographic and clinical parameters will provide objective measures of tumor activity and improve the accuracy of GBM tumor surveillance.

## 1. Introduction

Glioblastoma (GBM; astrocytoma grade IV) is the most common and severe manifestation of diffuse glioma. GBMs typically arise as primary *de novo IDH*-wildtype tumors and carry an exceedingly poor prognosis with a median survival of 15 months [1]. *IDH*-mutations confer a prognostic advantage to patients and are typical of less aggressive astrocytoma grade II-III tumors and their progressions to secondary GBM [2,3]. Current standard-of-care for GBM patients involves maximal debulking surgery followed by concomitant radiotherapy and temozolomide (TMZ) chemotherapy (STUPP protocol) [4]. Unfortunately, GBM recurrences are inevitable owing to the diffusely infiltrative and heterogeneous nature of tumors. Tumor growth and treatment responses are routinely monitored by radiographic and neurological assessments [5], however these surveillance methods cannot readily distinguish tumor progression from common treatment-related effects such as pseudo-progression, pseudo-regression and radionecrosis [6]. The inability to discern confounding treatment-related effects, particularly during progression, adds significant uncertainty to clinical management. Arguably, real advances in GBM patient care are unlikely without a sensitive means of tracking tumor activity.

Minimally invasive liquid biopsies offer an elegant and compelling solution to the inadequacies of current tumor monitoring approaches. Liquid biopsies sample and measure tumor-derived molecules from body fluids, such as the blood, to assess the active molecular state of a tumor. The development of a glioma liquid biopsy requires well-defined panels of accessible, stable and informative biomarkers capable of stratifying patients and distinguishing tumor progression and treatment resistance from treatment effects. Extracellular vesicles (EVs), 30–1000 nm membrane-bound particles secreted by all cell types, hold major promise here. EVs carry an array of lipids, DNA, RNA and protein that reflect the identity and molecular state of their cell-of-origin [7]. EVs play essential roles in glioma biology [8] and importantly cross the blood-brain-barrier into the circulation [9], making them excellent candidates for monitoring brain tumors *in situ* [8]. Suitably, neoplastic cells secrete more EVs [10,11] and therefore tumor-derived material may be detected above levels of potential ‘noise’ from other non-neoplastic sources. Moreover, screening the contents of circulating EVs could permit the heterogenetic molecular landscape of gliomas to be assessed *in toto* which is often not possible in small tissue biopsies and therefore allow active surveillance of tumor evolution [12]. Glioma-EV biomarkers that correlate with histological, neuroradiographic and clinical parameters may also be useful objective surrogate endpoints in clinical trial models, allowing protocols to be more dynamic and adaptive. 

Biomarker studies have mainly analyzed EVs sourced from GBM cells [13,14,15], blood [16,17,18] and CSF [19,20]. A pure and abundant source of *ex-vivo* glioma-EVs would undoubtedly expedite biomarker discovery and the translation of clinically-useful liquid biopsy assays. Cavitron ultrasonic surgical aspirate (CUSA) washings are a rich source of EVs sampled directly from tumor microenvironments, collected during tumor resection surgeries [15]. We have shown that CUSA-EV proteomes are capable of stratifying patients by glioma severity and can distinguish GBM from grade II-III glioma patients [15,21]. The small non-coding RNA (sncRNA) contents of these surgically captured EVs have not yet been described.

Multiple sncRNA have been detected in EVs, including regulatory transcripts such as microRNA (miRNA), small interfering RNA, p-element induced wimpy testis (PIWI)-interacting RNA (piRNA) and small nucleolar RNA [22,23]. miRNAs, 18–24 nucleotide single-stranded RNAs [24] are the most thoroughly characterized to date, with expression profiles curated across multiple sample types and disease states (i.e., http://bioinfo.life.hust.edu.cn/EVmiRNA) [25]. miRNAs are post-transcriptional regulators with a powerful and diverse capacity for controlling gene expression [26,27] and have wide-reaching functional effects, estimated to regulate approximately two-thirds of human protein-coding genes [28]. EV-cargoed miRNAs are exquisitely sensitive to specific disease states and have the capacity to discriminate closely related diseases [29,30,31,32]. For example, EV miRNA expression levels can distinguish metastatic and non-metastatic tumors [33], high- and low-grade tumors [34,35] and change in response to surgical tumor resection [36,37]. Systematic fluctuations in EV miRNAs across disease states, their stability and ready access from bodily fluids, makes them promising candidates for patient stratification and tumor monitoring. Likewise, piRNAs (24–32 nucleotides) also have diverse roles and immense potential as EV biomarkers. Whilst they are the largest class of sncRNAs [38], they are understudied in EVs. piRNAs have known important roles in transposon silencing, epigenetic regulation, genome rearrangement, germ stem-cell maintenance and oncogenesis [23,39]. piRNA expression varies significantly across different human somatic tissues and in cancer, piRNA expression profiles distinguish normal and tumor tissue [40] and correlate with clinical parameters [41], making them potentially useful as unique cancer-specific markers [39]. Mounting evidence shows that miRNAs are selectively packaged [42,43,44] and piRNAs enriched within EVs [45]. Their enclosure within EV phospholipid membranes protects them from endogenous nucleases [46,47], making EV-cargoed sncRNA more stable than free-circulating species. While miRNAs have been profiled in glioma-EVs [18,48] and shown to promote glioma transformation [49], drug resistance [50] and invasion [51], there are few reports of piRNAs [23].

A recent pilot study revealed a stable serum EV miRNA signature that could predict a preoperative GBM diagnosis with 91.7% accuracy [18]. A distinct, yet overlapping EV miRNA signature was also determined for glioma II-III patients suggesting that glioma subtyping may be possible via a blood test [18]. Here, EVs were purified from CUSA specimens captured during glioma resection surgeries and in-depth sncRNA profiles were assessed by unbiased NGS to identify miRNA and piRNA species capable of distinguishing highly aggressive GBMs (astrocytoma grade IV) from lower-grade astrocytoma (grade II-III) tumors. We then explored the levels of putative sncRNA markers in pre-operative patient sera to determine which sncRNA species may be reliably used as specific GBM-EV biomarkers in the circulation.

## 2. Results

### 2.1. Characterisation of Enriched CUSA-EVs 

Crude EV preparations isolated from GBM (astrocytoma grade IV; *n* = 12, Appendix A) and astrocytoma II-III (GII-III; *n* = 5, Appendix A) CUSA washings were enriched on discontinuous OptiPrep^TM^ gradients to reduce the presence of soluble and intracellular contaminants [52]. Captured EVs were characterized in accordance with criteria outlined by the ‘minimal information for studies of extracellular vesicles 2018′ (MISEV2018). CUSA-EVs largely settled across buoyant densities of 1.090, 1.100 and 1.109 g/mL (Figure 1A). Nanoparticle tracking analysis (NTA) revealed similar size distribution profiles for the GBM and GII-III enriched CUSA-EV populations, with four main small-EV population peaks observed at 75–85, 105–115, 155 and 205–215 nm (Figure 1B). The enriched CUSA-EVs displayed vesicular morphologies when visualized by TEM (Figure 1C). Tandem mass spectrometry (LC-MS/MS) analysis of the CUSA-EV proteomes identified a total of 2913 proteins across all samples and 1400 proteins in EVs isolated from >80% of glioma patients (99% confidence and ≥2 peptides; Appendix A). Of these, 809 proteins were common to all patients, including 77 of the top 100 EV proteins listed by Vesiclepedia (Figure 1D, Appendix A). Approximately half of proteins identified in GBM and GII-III CUSA-EVs were annotated to extracellular exosomes (51.73% and 56.36% of proteins for GBM and GII-III CUSA-EVs, respectively; *p* < 0.01; Figure 1E) and fewer than 10% of proteins to potentially contaminating endoplasmic reticulum or golgi apparatus cellular compartments (Figure 1E). In addition, multiple non-EV proteins that commonly isolate with EVs such as TGFβ1/2, IFNG, VEGFA, FGF1/2 and EGF [53], were not confidently detected in the CUSA-EVs (Appendix A). EV fractions (1.090–1.109 g/mL) were combined for unbiased sncRNA sequencing.

### 2.2. CUSA-EVs Are a Source of High-Quality Small RNAs Suitable for Biomarker Discovery

The quality and content of RNA isolated from the enriched CUSA-EV populations was assessed by an Agilent Bioanalyzer. In line with previous EV observations, the total RNA extracted from CUSA-EVs comprised a main RNA population sized < 200 nts, with no discernible 18S/28S ribosomal RNA populations (Appendix A) [54]. Generated cDNA libraries displayed a 175 bp peak, exhibiting successful sRNA size selection, 5’ and 3’ adapter ligation, reverse transcription and library amplification (Appendix A). FastQC checks of the sequenced sRNA libraries (*n* = 17) revealed high-quality and consistent sequencing data for all samples. A Phred quality score >30 was detected per read (Appendix A) and base (Appendix A) for all samples, while the number of reads, the percentage of unique and duplicate reads and the average GC content was consistent across all CUSA-EV specimens (Appendix A). On average, 17,620,146 ± 3,394,996 reads were sequenced for the CUSA-EV specimens (*n* = 17, Appendix A). An average of 23.4% of the total mappable reads were mapped to mature miRNAs, 31.6% to rRNA, 19.7% to tRNA, 2.9% to mRNA and 2.7% to other RNA subtypes (Appendix A, Appendix A). piRNAs represented a very small portion of the sRNA species in CUSA-EVs, with an average of 1.4% of all mappable counts across all glioma CUSA-EV specimens. 

### 2.3. Significantly Dysregulated CUSA-EV miRNAs between GBM and GII-III Exhibit Functional Associations to GBM Molecular Pathways

Overall, 1480 miRNA species, with average unique molecular identifier (UMI) counts >10, were identified in both GBM (*n* = 12) and GII-III CUSA EVs (*n* = 5) and were selected for DE analysis (Appendix A). A total of 212 miRNAs were DE in GBM CUSA-EVs relative to GII-III (*p* ≤ 0.05; Appendix A). Of these, 27 reached stringent significance thresholds (|FC|≥ 2, B-H corrected adj. *p* ≤ 0.05; Figure 2A, Appendix A) including 21 miRNA species with increased (GBM-high) and 6 miRNAs with decreased levels (GBM-low) in GBM CUSA-EVs relative to GII-III (Figure 2A). 

We explored biological and canonical pathways associated with the DE miRNAs (212 species, *p* ≤ 0.05; Appendix A) by performing a core expression analysis on predicted mRNA targets using Ingenuity^®^. mRNA targets of DE miRNAs were functionally linked to molecular mechanisms of cancer (*p* = 1.63 × 10^−45^; Figure 2B) and GBM signaling (*p* = 7.82 × 10^−27^), where identified miRNAs were predicted to alter the activity of 43 pathway molecules (Appendix A).

### 2.4. CUSA-EV piRNAs Distinguish GBM from GII-III 

Overall, 91 piRNA species were confidently identified in CUSA-EVs with an average UMI ≥ 10 in both GBM and GII-III cohorts (Appendix A). Of these, 10 piRNAs were significantly DE in GBM CUSA-EVs relative to GII-III (|FC| ≥ 2, B-H corrected adj. *p* ≤ 0.05, Table 1). All piRNA species had significantly higher expression in GBM CUSA-EVs (Table 1). Notably, hsa_piR_16659, 016658, 020365, 004153, 020829 and 016677 species showed higher than 5-fold increases in GBM CUSA-EVs and all significant piRNAs are reported as cancer-associated molecules in the literature (Table 1).

### 2.5. Identified CUSA-EV sncRNAs Can Discriminate Serum-EV profiles of GBM and GII-III Patients and Healthy Controls

We then determined whether the miRNA and piRNA species identified in CUSA-EVs (UMI ≥ 10) were also present and similarly expressed in EVs captured from patient and healthy control (HC) sera. sncRNAs were confidently identified in serum-EVs if they had an average read count ≥ 5 in at least 2 sample groups (GBM [*n* = 12], GII-III [*n* = 10] and HC [*n* = 16]; Appendix A). The expression of 410 miRNAs (Figure 3A; Appendix A) and 64 piRNAs (Figure 4A; Appendix A) identified in CUSA-EVs were detectable in serum-EVs. 

We next tested whether the overlapping miRNA species could be used to discriminate serum-EV profiles from GBM and GII-III patients and HCs. Of the 410 miRNAs common to EVs isolated from both sources, 143 were DE in serum-EVs across the three sample groups (pairwise comparisons, adj. *p* ≤ 0.05; Figure 3A, Appendix A). Using PCA, we assessed whether the expression levels of the overlapping significant DE 143 miRNAs (Figure 3B) could distinguish the serum-EV samples into their respective cohort/diagnostic group. The serum-EV profiles were observed to cluster according to their respective cohort grouping, with the more aggressive GBMs clustering to the far right of the PCA, HCs to the left and GII-III intermedially dispersed (Figure 3B).

Encouragingly, 14 of the 27 DE miRNAs between GBM and GII-III CUSA-EVs (adj. *p* ≤ 0.05, Figure 2A) were also significantly expressed in serum-EVs (*p* ≤ 0.05; Figure 3C). These significant overlapping miRNAs included previously reported DE miRNAs in serum-EVs from GBM, GII-III and corresponding age- and gender-matched HCs (adj. *p* ≤ 0.05; 106b-3p, 378a-3p, 21-5p, 486-3p) [18]. Overlapping miRNAs with borderline significance in GBM CUSA-EVs relative to GII-III included miR-335-3p, 134-5p, 370-3p, 185-5p, 25-3p, 409-3p and 382-5p (0.05 ≤ adj. *p* ≤ 0.1;). Analogous expression of miR-486-3p in CUSA (Figure 5A,B) and serum-EVs (Figure 5C) was observed, with higher levels in GBM, relative to GII-III. An miRBase word cloud [62] was generated for hsa-miR-486 and lists functional information extracted from 182 open access papers and 1080 associated sentences (Figure 5D). Prominent keywords include ‘cancer,’ ‘increased’ and ‘plasma’ as well as ‘staging,’ ‘survival’ and ‘prognosis’ (Figure 5D). Interestingly, miR-106b-3p and miR-378a-3p were both more abundant in GBM CUSA-EVs relative to GII-III, however, in serum, these species were significantly higher in both GBM and GII-III serum-EVs relative to HCs (Figure 3C). 

Similarly, we then determined whether the overlapping piRNA species could be used to discriminate serum-EV profiles from GBM and GII-III patients and HCs using Oasis2.0. Of the 64 piRNAs common to EVs isolated from both sources, 33 were DE in serum-EVs across the three sample groups (pairwise comparisons, adj. *p* ≤ 0.05; Figure 4A, Appendix A). Using PCA, we assessed whether the expression levels of the overlapping significant DE 33 piRNAs could distinguish between serum-EV samples from different cohort/diagnostic groups (Figure 4B). While serum-EV profiles from glioma patient (GBM and GII-III) sera clustered away from HC sera, DE piRNAs were unable to discern the GBM and GII-III patients. 

Four piRNA species (piR_016658, 016659, 020829 and 20490) with significantly higher expression in GBM CUSA EVs relative to GII-III (adj. *p* ≤ 0.05, Table 1) also changed between glioma and HC sera (*p* ≤ 0.05; GBM vs. HC and GII-III vs. HC; Figure 4A). This included piRNA species, 016658, 016659 and 020829 with strikingly higher expression levels (10.6-23.7-fold in GBM CUSA-EVs; Table 1). The distribution of piR_016658, 016659, 020829 and 20490 expression across the serum-EV samples are visualized in box plots (Figure 4C). While piR_016658, 015569, 020829 and 20490 had significantly higher expression in GBM relative to GII-III CUSA-EVs, piR_020829 and 020490 displayed higher levels and piR_016658 and 015569 lower levels in GBM and GII-III sera-EVs, relative to the HCs (Appendix A). While these significant piRNA species could discriminate glioma sera-EVs from HCs, they were unable to distinguish the sera of more aggressive GBM patients from GII-III. This is further substantiated by the PCA analysis (Figure 4B) based on data from 33 overlapping DE piRNA species across CUSA and serum-EVs, where GBM and GII-III samples could not be distinguished based on their piRNA expression. While there were no significant distinguishing piRNA species identified between GBM and GII-III serum-EVs, a trend was observed between the magnitude of piR_016658 and piR_016659 changes and glioma severity (Figure 4C-1,C-2); where HCs showed the highest expressions of piR_016658 (Figure 4C-2) and piR_016659 (Figure 4C-1), reduced levels were measured in GII-III patients and greater reductions observed in the GBM cohort. 

## 3. Discussion

Accurate and sensitive glioma blood-based biomarkers will arm clinicians with much needed information to offer patients definitive and timely answers about the active state of their cancer and the effectiveness of their treatment. Previously, we found neurosurgical CUSA washings to be a viable, rich source of glioma-derived EVs suitable for comprehensive proteomic studies [15,21]. Here, we employed a density gradient ultracentrifugation method to enrich small EV subtypes (<200 nm), as described before [21] and show that CUSA-EVs are a source of high quality sncRNA for biomarker discovery by high-throughput unbiased NGS.

### 3.1. Prominent miRNA Markers in Surgically Captured Glioma EVs

Hallmark characteristics of GBM tumors, including high proliferative rates, diffuse cell invasion, microvascular proliferation and necrosis are underpinned by complex, heterotypic interactions within tumor microenvironments. Here, EVs are known important modulators of intercellular communication, casting influence over recipient cells via the exchange of biomolecules, including miRNAs, to induce genetic alterations and dysregulate molecular signaling pathways. We employed unbiased NGS and confidently detected 1480 miRNA species in CUSA-EVs captured directly from glioma tumor microenvironments. Functional pathway analysis of mRNAs targeted by CUSA-EV miRNAs (212 species DE in GBM relative to GII-III; *p* ≤ 0.05) showed clear associations to the GBM signaling pathway (Appendix A) and more broadly, ‘molecular mechanisms of cancer.’ Significant links to IL-6 inflammatory pathways and p53 signaling which play important roles in gliomagenesis [63,64,65,66] were also observed, as well as functional associations to tumor morphology, cell growth, proliferation and cellular movement that are pertinent features of highly infiltrative and aggressive GBMs. Of these miRNA species, 27 were DE in GBM CUSA-EVs relative to GII-III patients at the stringent significance threshold (|FC| ≥ 2, B-H corrected adj. *p* ≤ 0.05). Interestingly, miRNA species with the highest fold changes in GBM CUSA-EVs include miR-21-5p (20.66-fold increase), 550a-3p (12.56-fold decrease), 335-5p (10.49-fold increase), 146b-5p (8.36-fold increase) and 155-5p (8.17-fold increase), all of which have reported tumor-supportive roles [67,68,69,70,71,72,73,74].

We identified multiple DE CUSA-EV miRNAs that directly interact with GBM signaling pathways (Appendix A). miR-21-5p is commonly overexpressed in tumors, acting as an oncogene [50,68,75] and promoting tumor progression by targeting phosphatase and tensin homolog (*PTEN*) [68], the most commonly lost or depleted gene implicated in primary brain tumors [76]. Multiple studies have reported miR-21 as a biomarker in GBM-EVs [8,20,48,77,78,79]; higher miR-21-5p levels were detected here in CUSA-EVs harvested directly from the GBM microenvironment, which is analogous to reports of miR-21-5p expression in GBM tissue [80]. In serum-EVs, however, miR-21-5p was significantly lower in both GBM and GII-III relative to HCs (Figure 3C). The discordance in miR-21-5p levels in serum-EVs might reflect the highly heterogenous sources of EVs in serum and the systemic effects of glioma pathology and/or treatment. 

Interestingly, miR-22-3p (4.9-fold higher in GBM CUSA-EVs) was predicted to increase in response to changes in p53 activity in the GBM signaling pathway (Appendix A). miR-22-3p is a p53-activated miRNA [81] and functions to regulate cell proliferation and apoptosis [82]. Typically, p53 expression is deregulated in GBM [83] and increased miR-22-3p is observed in malignant astrocytoma tissues [84] and other cancers [82,85,86].

### 3.2. CUSA-EV Findings Substantiate miR-486-3p, miR-106b-3p, miR-378a-3p, miR-185-5p and miR-25-3p as Serum-EV Markers for GBM/Glioma

In our previous study, peripheral miRNA EV signatures were determined for *IDH-wt* GBM and *IDH-mut* glioma II-III patients pre-operatively, when tumor loads are at their maximal burden [18]. The serum-EV miRNA sequencing data was analyzed by machine learning approaches to determine whether miRNA signatures could discern GBM and GII-III patients from age-gender matched HC participants [18]. Seven miRNA species were determined to be the most stable classifiers for GBM (miR-182-5p, 382-3p, 339-5p, 340-5p, 485-3p, 486-3p and 543) with an overall predictive power of 91.7%; within multivariate models, six iterations of these markers were capable of distinguishing GBM patients from HC with 100% accuracy. The study also found a partially overlapping and distinct signature for GII-III, with miR-7d-3p, 106b-3p, 130b-5p, 185-5p and 98-5p determined as stable classifiers for GII-III [18]. For this study, CUSA washings collected at a near identical time-point in glioma patient clinical management, yielded a rich source of glioma-EVs captured directly from the tumor microenvironments. These were compared to the previously published peripheral serum-EV sncRNA data from GBM, GII-III and HC samples [18]. The serum-EV sncRNA data was re-processed using the Oasis2.0 pipeline and PCA plots were generated for overlapping sncRNAs in CUSA and serum-EVs, to determine their discriminatory capabilities. 

Significantly high miR-486-3p levels were observed in both CUSA and serum-EVs from GBM patients (Figure 5). In our previous multivariate analysis, miR-486-3p was identified as one of the top four features that distinguish GBM from GII-III patient sera [18]. miR-486-3p is overexpressed in primary glioma tissues and its expression is strongly correlated with glioma grade and poor overall survival [87]. Moreover, increased miR-486-3p levels were linked to NF-κB activation which promotes glioma aggression [87]. The *miR-486* gene is located in the intron of the *sAnk1* gene [88]; the *Ank1* promoter region contains TGF-β-responsive element (SRE) and tandem repeats of the TCF/LEF transcriptional responsive element, connecting miR-486 regulation with TGF-β/Smad and Wnt/β-catenin signaling [87]. *IDH1*-mutations, present in the GII-III cohort here, were shown to negatively regulate Wnt/β-catenin signaling [89] and TGF-β expression is lower in *IDH*-mutant compared to *IDH*-wildtype gliomas [90]. Thus, in addition to its potential use as a marker for glioma aggression, miR-486 levels may correspond to *IDH*-mutational status. miR-486-3p is also a marker for other aggressive tumor types, with high expression observed in oral squamous cell carcinoma tissue [91], in the CSF of medulloblastoma patients [92] and in colorectal cancer [93]. Paradoxically, other studies have reported that increased miR-486-3p expression promotes chemo-sensitivity of GBM to TMZ by targeting MGMT [94] and esophageal carcinoma cell-lines to 5-fluorouracil [95], suggesting important roles for miR-486-3p in targeting cell survival molecular pathways in response to cytotoxic stimuli. The aberrant expression of miR-486-3p across multiple pathologies and in response to chemotherapy makes it an attractive biomarker for monitoring of glioma progression and treatment responses [95]. Further study using larger and more homogenous glioma cohorts including specimens taken during recurrent progression will confirm whether miR-486-3p is a useful marker for glioma severity and progression.

Analysis of the predicted targets of these deregulated miRNAs suggests that these miRNAs are, in fact, impacting on functions relevant for the cellular response to a cytotoxic stimulus. These data highlight the identified miRNAs as potential candidates for clinical response monitoring of chemotherapy and provide a basis for future studies to determine the effect of chemotherapy on the expression in vitro and *in vivo*. Finally, further investigations are warranted to identify the cellular pathways which are affected by the deregulated miRNAs, suggests that these miRNAs may target molecular pathways involved in cell survival after chemotherapy.

Further overlap between the NGS analyses of serum-EVs and CUSA-EVs also substantiates miR-106b-3p, miR-378a-3p, miR-185-5p and miR-25-3p as potential blood based-markers for glioma detection. Comparably, miR-106b-3p and 185-5p were previously identified as stable classifiers for GII-III and also found overlapping significance between GBM and HC [18]. With further study and validation, these markers may be associated with tumor burden and thus useful for detecting tumor recurrences. As miR-486-3p and miR-25-3p displayed significantly higher expression in more aggressive GBM-EV specimens, these markers in particular should be further studied in the context of low-grade gliomas patients who are placed on an active surveillance regimen in the knowledge that progression to a higher grade tumor is likely.

### 3.3. Cancer-Associated piRNAs in EVs can Distinguish Glioma Patients

piRNAs (20-30 nt) are endogenous sncRNAs known as the ‘guardians of the genome’ that have been previously reported in EVs [96]. They are unique to mammals and have silencing functions to protect genes from invasive transposable elements by repressing transposons post-transcriptionally [97,98]. Among other silencing RNAs, piRNAs are less studied because of their complexity and uniqueness [97]. Here, we show that while piRNAs make up the smallest portion of mappable sncRNA reads within CUSA-EVs (Appendix A), they are capable of distinguishing GBM CUSA-EVs from GII-III (Table 1). Multiple significant DE piRNAs identified in glioma CUSA-EVs are aberrantly expressed across multiple human cancers, including breast [55], endometrial [57] and brain cancer [61,99]. In particular, piR_016658, 016659 and 020829 had the highest fold-expression in GBM CUSA-EVs as well as significant DE in the serum-EVs of GBM and GII-III patients (Figure 4C). In plasma-EVs, piRNAs were shown to represent 1.31% of all mappable counts [54] and have been detected at high levels in EVs from cancer patients [100]. A few studies have investigated piRNAs in GBM specimens [61,101] and interestingly, piR_020490, _018573, _020548 and _001318 were detected in higher levels in GBM tissues compared to healthy brain tissue [61]. However, while many molecular profiling studies have sequenced the small RNA content of glioma-EVs, little is known about glioma-EV piRNAs. To the best of our knowledge, this is the first study that has investigated piRNAs in glioma-EVs and compared their expression across EVs captured from patients with different glioma subtypes and WHO grades.

### 3.4. Study Design Considerations

While there are multiple subtypes of sncRNAs, this study focused on miRNA and piRNA due to the expanding knowledge of their biological functions. The high-discovery power of NGS allowed for reliable and reproducible analysis of the RNA with little artefacts. However, sensitive methods such as NGS easily detect sample contaminants. To circumvent this issue, CUSA-EVs were pre-treated with RNase prior to RNA extraction to remove co-isolating extravesicular RNAs that might otherwise obscure selective and specific assessment of RNAs cargoed in CUSA-EVs. Bioanalyzer traces revealed that the CUSA-EV RNAs contained a main population below 200 nt and almost no detectable ribosomal RNA (Appendix A), in line with previous reports for EVs [42,44]. Approximately half of the sncRNA contents of CUSA-EVs were mappable and predominantly comprised of rRNAs (31.6%) and miRNAs (23.4%). These results conform with other studies that report more than half of the EV RNA content is comprised of rRNA and tRNA, while miRNA and piRNA sequences comprise a relatively lower and variable portion of the small RNA within EVs [45]. 

CUSA fluid is captured during surgical resection, a critical time point for biomarker discovery as is it the only time at which the glioma microenvironment is directly accessible [21]. In this study, density gradient ultracentrifugation enriched for small-EV subtypes from the CUSA fluid. While the presence of larger EV subtypes in our preparations cannot be discounted, there are currently no reported molecular markers that categories EVs into small/large subtypes [53]. It is proposed that smaller-EVs are depleted from nuclear, golgi apparatus and endoplasmic reticulum proteins relative to the cell [53]. Here, FunRich annotated approximately 50% of the CUSA-EV proteins to exosomes and fewer than 10% to the endoplasmic reticulum or golgi apparatus (Figure 1E). Our observations here expand the value of CUSA washings for EV biomarker discovery and profiles of EV-RNAs associated with glioma subtypes and clinical parameters will aid the development of targeted GBM liquid biopsy assays.

sncRNA species associated with more aggressive astrocytoma phenotypes were identified by comparative analysis of CUSA-EVs from *IDH-wt* GBM (GBM) and *IDH-mut* grade II-III astrocytomas (GII-III). However, sample limitations must be acknowledged. Due to the low incidence rate and prospective collection of CUSA fluid, the GII-III cohort is comprised of a small number of *IDH*-*mut* astrocytoma patients (*n* = 5). While *IDH-mut* grade II-III astrocytoma can progress to secondary GBM, they are molecularly distinct entities. It is important to note that the serum-EV GII-III cohort also includes *IDH-mut* oligodendroglioma patients. Moreover, the groups were comprised of both treatment-naïve patents as well as those with recurrent tumor progression. To resolve markers associated with treatment resistance and recurrence, comparisons of more homogenous groups at similar treatment stages is important. The patient cohorts studied do, however, provide insight into glioma-EV populations and EV-sncRNA profiles associated with more aggressive and invasive glioma phenotypes. Expansion of this discovery analysis to include larger and well-defined cohorts of glioma subtypes will likely resolve more specific and nuanced biomarkers for glioma patient stratification. 

### 3.5. Discrepancies between Serum and CUSA-EV RNA Expression Patterns 

While multiple serum-EV miRNA and piRNA markers showed significant and similar expression within CUSA-EVs, there were discrepancies observed in both miRNA and piRNA expression patterns. In serum-EVs, multiple miRNA and piRNAs were present in lower levels in GBM patients and GII-III compared to HCs, while in CUSA-EVs, the same RNA species were expressed at higher levels in GBM relative to GII-III. Due to the study comparison limitations described above, there are additional stark differences that may account for these contradictory findings. First, the serum-EV study investigated the total EV population within the blood, which would have been composed of EVs derived from all cell-types of the body. The sncRNA constituents of serum-EVs would be reflective of the total systemic effects related to tumor biology and treatment, rather than reflect a specific glioma-EV molecular signature. Moreover, EVs from CUSA and serum were isolated by different methods; CUSA-EVs were isolated and enriched by density gradient ultracentrifugation and serum-EVs were isolated by SEC. Therefore, enrichment of different EV subpopulations and differences in RNA profiles due to isolation methodology cannot be discounted [42,53,102]. Further investigations incorporating both targeted and holistic approaches are necessary to validate potential circulating EV miRNA and piRNA biomarkers in glioma patient populations. 

## 4. Materials and Methods 

### 4.1. Neurosurgical Aspirate Specimens 

Cavitron ultrasonic surgical aspirate (CUSA) washings were collected from astrocytoma II-IV patients prospectively over a 36-month period (October 2015–2018). Informed consent was obtained from all participants for biobanking of surgical specimens by the Sydney Brain Tumor Bank, SLHD HREC protocol X19-0010. Specimens were processed and analyzed under the University of Sydney HREC approval numbers 2012/1684 and 2019/705 (approved on 5th September, 2019). Diagnoses were confirmed histopathologically and included primary WHO *IDH*-wildtype GBM (astrocytoma grade IV; *n* = 12) and *IDH*-mutant astrocytoma grade II-III (GII-III; *n* = 5). A summary of the glioma patient cohorts are provided in Appendix A.

### 4.2. Processing CUSA Washings and EV Enrichment

Tissue fragments were strained from the CUSA washings and formalin-fixed for routine diagnostic histopathology at RPA Tissue Pathology and Diagnostic Oncology Department. The CUSA filtrate was recovered and suspended cells were isolated by centrifugation (350× *g*, 10 min, 4 °C; viability >98%). The CUSA supernatant was clarified from cell debris and larger particulates by centrifugation (2000× *g*, 20 min, 4 °C) and 0.22 µm filtration. Clarified CUSA fluid was stored at −80 °C.

EVs were pelleted from CUSA fluid by ultracentrifugation (100,000× *g*, 16 h, 4 °C, SW41 Ti/45 Ti, Beckman Coulter L80XP) and washed with sterile-filtered ice-cold PBS (100,000× *g*, 2 h, 4 °C). Crude CUSA-EV pellets were resuspended in PBS, the protein content estimated by Qubit^®^2.0 fluorometric quantitation and aliquots stored at −80 °C. Crude CUSA-EV preparations were further purified and enriched by density gradient ultracentrifugation using Optiprep^TM^ (60% (*w*/*v*) aqueous iodixanol; Axis-Shield PoC, Norway). Working solutions of 40% (*w*/*v*), 20% (*w*/*v*), 10% (*w*/*v*) and 5% (*w*/*v*) OptiPrep^TM^ were prepared in 0.25 M sucrose/10 mM Tris, pH 7.5. From the base of a 13 mL ultraclear tube (14 × 89 mm; Beckman Coulter, Cat No. 344059), 3 mL of 40% (*w*/*v*), 20% (*w*/*v*) and 10% (*w*/*v*) OptiPrep^TM^ working solutions were carefully layered, followed by 2.5 mL of 5% (*w*/*v*) solution. Crude CUSA-EVs (300 µg) diluted in 500 μL of 0.25 M/10 mM Tris, pH 7.5, were layered on top of the gradient. Gradients were ultracentrifuged (100,000× *g*, 18 h, 4 °C, acc 1, no brake; SW41 Ti, Beckman Coulter L80XP) and 12 × 1 mL fractions of increasing density were collected manually with a cut-off pipette tip. A control, blank gradient was run in parallel and each fraction was measured on a sensitive analytical balance to determine fraction density. Fractions were washed with 12 mL of PBS and ultracentrifuged (100,000× *g*, 3 h, 4 °C). Fractionated CUSA-EV pellets were resuspended in 50 μL PBS and stored at −80 °C.

### 4.3. EV Characterisation 

The size distributions and concentrations of EV populations were measured by NTA software (version 3.0) in triplicate using the NanoSight LM10-HS (NanoSight Ltd, Amesbury, UK). The NTA system was configured with a tuned 532 nm laser and a digital camera system (CMOS trigger camera). EVs were diluted with sterile-filtered PBS (viscosity 1.09 cP) to ensure 20-100 particles were detected within the field of view of the NanoSight CCD camera. The NTA software captured 60 s video recordings of the EVs in triplicate, at 25 frames per second with default minimal expected particle size, minimum track length and blur setting. The temperature of the laser unit was controlled to 25 °C. The videos were analyzed by NTA3.0, which translated the Brownian motion and light scatter properties of each individual laser-illuminated particle into a size distribution (ranging from 10 to 1000 nm) and concentration (particles per mm) while simultaneously calculating their diameter using statistical methods [103]. The EV size distributions and concentrations were analyzed in Excel (Microsoft). The morphology and size of EVs were also assessed by transmission electron microscopy (TEM). EVs were re-suspended in dH_2_O, loaded onto carbon-coated, 200 mesh Cu formvar grids (ProSciTech Pty Ltd, QLD, Australia; Cat No. GSCU200C) and fixed with 2.5% glutaraldehyde in 0.1 M phosphate buffer (pH 7.4). Samples were negatively stained with 2% uranyl acetate for 2 min, dried at RT for 3 h and then visualized at 40,000×, 80,000× and 100,000× magnification on a Philips CM10 Biofilter TEM (FEI Company, OR, USA) equipped with an AMT camera system (Advanced Microscopy Techniques, Corp., MA, USA) at an acceleration voltage of 80–120 kV. To confirm the presence of canonical EV marker proteins, the EV proteomes were prepared and analyzed by LC-MS/MS and annotated as detailed before [15,21]. 

### 4.4. RNA Extraction and Quality Control

EVs were resuspended with PBS to a total volume of 1 mL and pre-treated with 3.5 U (50 µg/µL) RNase A (10 min, 37 °C) to digest any co-isolated extravesicular RNA. Total RNA was extracted from the EVs using the plasma/serum circulating and exosomal RNA purification kit (Slurry Format; Norgen Biotek Corp, ON, Canada; Cat No. 42800) as per manufacturer’s instructions. RNA samples were concentrated with an Eppendorf 5301 Concentrator (Eppendorf, AG, Hamburg) and stored at −80 °C. RNA quality, integrity and concentration (pg/µL) were assessed with a eukaryotic RNA Picochip assay on an Agilent 2100 Bioanalyzer (Agilent Technologies, Santa Clara, CA) as per manufacturer’s instructions. As EVs contain limited 28S and 18S ribosomal RNA (rRNA) [54], an RNA integrity number (RIN) could not be assigned to each sample. However, the Agilent Bioanalyzer was useful to measure RNA concentration and nucleotide length.

### 4.5. Small RNA Library and Next Generation Sequencing (NGS)

Starting total template CUSA-EV RNA (4 ng) was used to prepare the small RNA (sRNA) libraries using a QIASeq^TM^ miRNA Library Kit (Qiagen, Venlo, Netherlands) [104]. The sRNA library was prepared, quality control and NGS was performed by the Ramaciotti Centre for Genomics, University of New South Wales. NGS was performed on an Illumina^®^ NextSeq^TM^ 500 NGS System (Illumina, San Diego, CA, USA). The libraries were sequenced in a single-end 75 bp, high-output protocol to generate an output of up to 400 million reads at a Q30 > 85%. 

### 4.6. Data Availability

CUSA-EV raw data are accessible at NCBI Gene Expression Omnibus (GEO accession no. GSE151547). Serum-EV raw data was accessed from NCBI Gene Expression Omnibus (GEO accession no. GSE122488) and is associated with previously published data [18]. The serum-EV cohorts used in this study are detailed in Appendix A. The methods for serum processing, EV isolation, EV characterization, sncRNA preparation and sequencing are previously described [18].

### 4.7. Data Pre-Processing, Normalisation and Differential Expression Analysis

Four fastq.files generated by NGS were merged for each sample and the quality of the sequencing data was assessed by FastQC (version 0.11.8; Babraham bioinformatics, UK). FastQC performed high-throughput sequence quality control, determined average read quality (*Q*-score) and base quality. Samples with a *Q*-score above 30 were considered high-quality data and used for the remainder of the analysis. To map the CUSA-EV sRNAs, the fastq files were uploaded to the Qiagen GeneGlobe^®^ Data Analysis Centre for analysis by the QiaSeq^®^ miRNA primary quantification pipeline (Qiagen, Venlo, Netherlands). Briefly, the 3′ adapter and low-quality bases were trimmed from the reads using Cutadapt [105], insert sequences and UMI sequences were identified and annotated and Bowtie was used to remove known contaminants by sequentially aligning the reads [106]. Reads were mapped by perfect match to miRBase mature, miRBase hairpin, noncoding RNA, mRNA and other RNA. At each step, only unmapped sequences passed to the next step and remaining unmapped sequences were mapped to the human miRBase mature database (human: Genome Reference Consortium GRCh38) where up to two mismatches were tolerated. Read and UMI counts were exported to Excel^®^. The percentage of mapped reads is provided in Appendix A, while raw UMI counts and RNA reads are provided in Appendix A. The QiaSeq^®^ miRNA secondary analysis pipeline was used to perform differential-expression (DE) analysis of CUSA-EV sRNAs. Raw sRNA UMI counts were normalized using EdgeR trimmed mean of M-values (TMM) and sRNA with UMI counts ≥ 10 in both GBM and GII-III cohorts were selected for DE analysis. The *p*-values were adjusted with the Benjamini-Hochberg (B-H) correction method at a false discovery rate (FDR) threshold of 5%. 

Serum-EV sRNA sequencing data from pre-operative glioma patients (cohort summary provided in Appendix A) were downloaded from NCBI Gene Expression Omnibus (GEO accession no. GSE122488). The data was trimmed, reads aligned to the human genome and normalized read counts were generated by Oasis2.0 and DE analysis was performed as described above. 

### 4.8. Data Visualisation and Ingenuity Pathway Analysis

Ingenuity^®^ Pathway analysis software was used to explore and measure biological/functional associations to the CUSA-EV miRNAs (IPA; Ingenuity Systems, USA). miRNA target filters were applied to lists of DE miRNAs (significance thresholds set at unadjusted *p* ≤ 0.05) and mRNA target lists were generated based on highly-predicted or experimentally-observed confidence levels. Core expression analyses were performed using default criteria to determine the most significant biological and canonical pathways targeted by the dysregulated miRNAs. Using IPA’s ‘grow tool’ constrained to experimentally-observed, direct interactions in humans, DE miRNAs were linked to their targets within the GBM signaling pathway. Principal component analysis (PCA) plots were generated in BioVinci (BioTuring, v1.1.5) and boxplots of DE sRNAs were generated with OriginPro 2020 (Origin Lab Corporation).

## 5. Conclusions

In this study, unbiased NGS was used to characterize EV-cargoed sncRNA species captured directly from the tumor microenvironments of GBM and grade II-III astrocytoma patients. Significant changes in miRNA and piRNA expression were found in GBM-EVs compared to GII-III EVs, several of which play important roles in gliomagenesis and were previously reported as GBM peripheral serum-EV biomarkers. Our results show that CUSA-EV miRNAs can elucidate and/or validate potential prognostic biomarkers for GBM. With further study, these targets could offer avenues for tumor staging and/or disease progression monitoring via peripheral blood sampling of glioma-EVs.

## Figures and Tables

**Figure 1 ijms-21-04954-f001:**
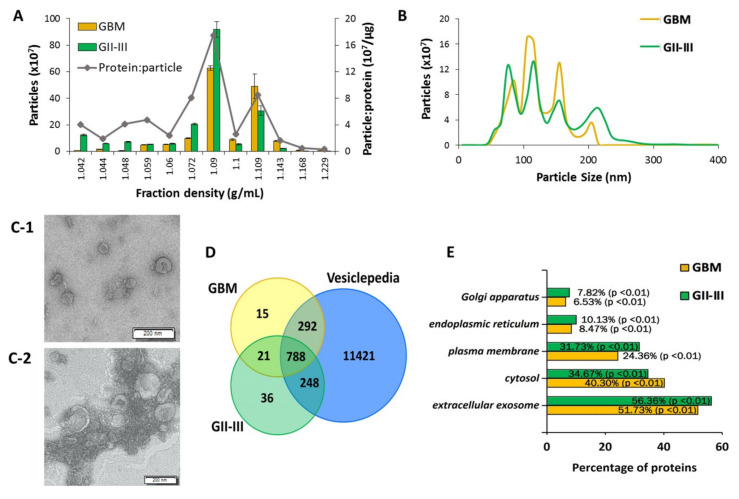
Characterization of extracellular vesicles (EVs) isolated from glioblastoma (GBM) and astrocytoma GII-III surgical aspirates. EVs isolated from cavitron ultrasonic surgical aspirate (CUSA) washings from astrocytoma II-IV patients were enriched on OptiPrep^TM^ density gradients. Nanoparticle tracking determined (**A**) particle numbers measured in each of the 12 density fractions collected (error bars indicate the standard error of the mean) and average particle to protein ratio and (**B**) size distributions of enriched GBM and GII-III CUSA-EVs in combined fractions 7–9. (**C-1**) Transmission electron microscopy (TEM) was used to visualize enriched CUSA-EVs from GBM and (**C-2**) astrocytoma grade II surgeries, scale bar = 200 nm. (**D**) Overlap of liquid chromatography tandem mass spectrometry (LC-MS/MS) sequenced proteins confidently identified in at least 80% of GBM (yellow) and GII-III (green) CUSA-EVs with proteins listed in Vesiclepedia (blue) associated with ‘extracellular vesicle,’ ‘exosome,’ ‘microparticle’ and ‘microvesicle.’ (**E**) FunRich annotations of GBM and GII-III CUSA-EV proteins to cellular compartments.

**Figure 2 ijms-21-04954-f002:**
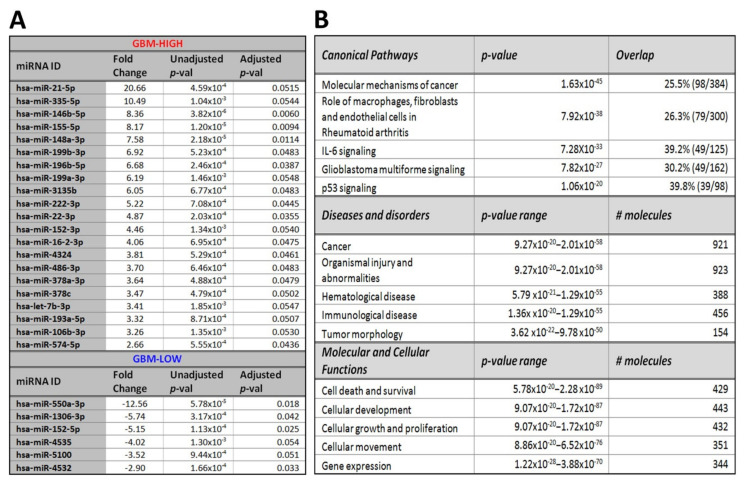
Significant miRNA species in GBM relative to astrocytoma GII-III CUSA-EVs. (**A**) Significant differential-expression (DE) miRNAs in enriched CUSA-EVs captured during astrocytoma debulking surgeries. miRNAs with significantly high (GBM-high; (FC ≥ 2, B-H adj. *p* ≤ 0.05) and low (GBM-low; FC ≤ −2, B-H adj. *p* ≤ 0.05) expression in GBM relative to GII-III CUSA-EVs are listed. (**B**) Functional pathway analysis of mRNAs targeted by 212 changing miRNAs (*p* ≤ 0.05) in GBM CUSA-EVs. The top canonical pathways, diseases and disorders and molecular and cellular functions are listed with the number of overlapping molecules and significance associations (right-tailed Fisher exact test, *p*-value).

**Figure 3 ijms-21-04954-f003:**
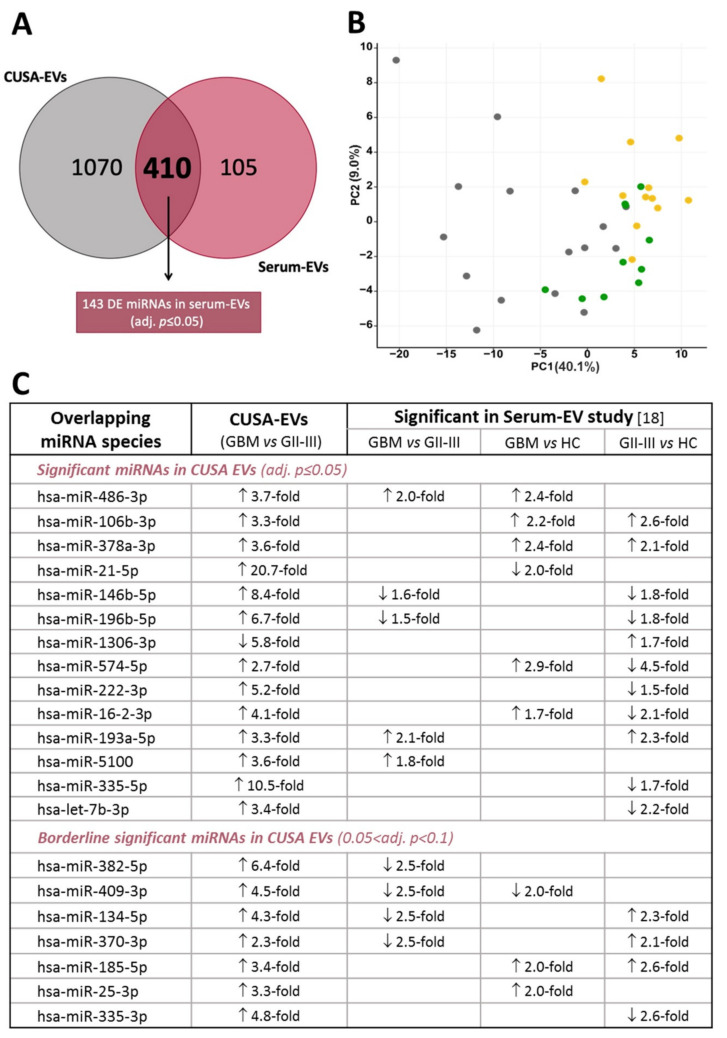
miRNA species in glioma-EVs captured from both CUSA fluid and serum. (**A**) Venn diagram showing the overlap of miRNAs detected in CUSA-EVs (average UMI count ≥ 10 in both GBM and GII-III) and serum-EVs (average read count ≥ 5 in at least two of three cohorts [GBM, GII-III and healthy controls; HC]). The Oasis 2.0 analytical pipeline was used to perform differential expression analyses and identified 143 significant miRNAs in serum-EVs (adjusted *p* ≤ 0.05; pairwise comparisons). (**B**) Principal component analysis plot shows the clustering pattern for GBM (yellow, *n* = 12), GII-III (green, *n* = 10) and HC (grey, *n* = 16) in serum-EVs, based on the expression of 143 differentially expressed overlapping miRNAs; the proportion of variance is explained by PC1 (*x*-axis) = 40.1% and PC2 (*y*-axis) = 9.0%. (**C**) Summary table of overlapping significant DE miRNAs in both CUSA-EVs (GBM vs. GII-IIIl) and serum-EVs (pairwise comparisons). Arrows depict direction of miRNA fold changes.

**Figure 4 ijms-21-04954-f004:**
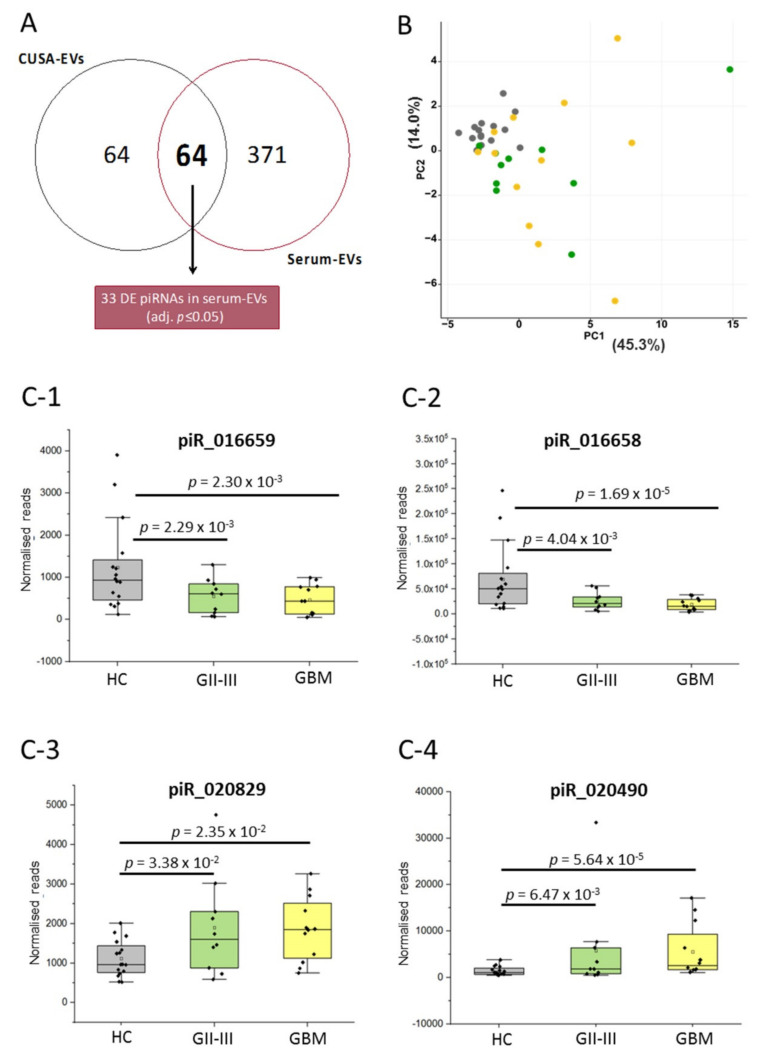
piRNA species in glioma EVs captured from CUSA fluid and serum. (**A**) Venn diagram showing the overlap of piRNAs detected in CUSA-EVs (average UMI count ≥ 10 in both GBM and GII-III) and serum-EVs (average read count ≥ 5 in at least two of three cohorts [GBM, GII-III and healthy controls; HC]). The Oasis 2.0 analytical pipeline was used to perform differential expression analyses and identified 33 significant piRNAs in serum-EVs (adjusted *p* ≤ 0.05; pairwise comparisons). (**B**) PCA plot shows the ability of 33 significant piRNA species to cluster serum-EV profiles from GBM (*yellow, n* = 12), GII-III (*green*, *n* = 10) and HC (*grey*, *n* = 16) specimens; the proportion of the variance is explained by PC1 (*x*-axis) = 45.3% and PC2 (*y*-axis) = 14.0%. (**c**) Significant CUSA-EV piRNAs are also significant in circulating serum-EVs. The box-plots show (**C-1**) piR_016659, (**C-2**) piR_016658, (**C-3**) piR_020829 and (**C-4**) piR_020490 serum-EV expression levels for HCs (*n* = 16, grey), GII-III (*n* = 10, green) and GBM (*n* = 12, yellow). The normalized piRNA reads for each individual patient and denoted by a black dot. The upper error bars signify the 90^th^ percentile and lower error bars represent the 10th percentile, the middle line represents the median and the open square signifies the mean.

**Figure 5 ijms-21-04954-f005:**
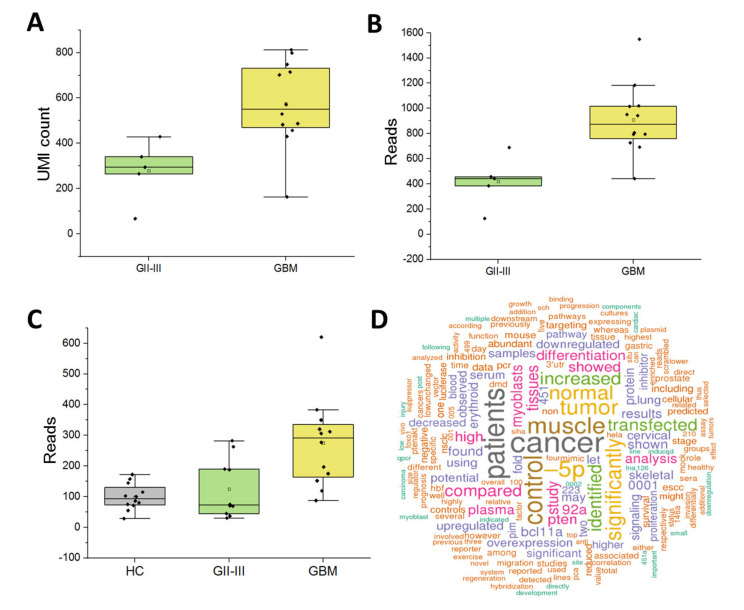
Expression of putative biomarker miR-486-3p across glioma cohorts. Box-plots show miR-486-3p expression in CUSA-EVs as (**A**) unique molecular identifier (UMI) counts and (**B)** reads from GBM and GII-III patients. Number of miR-486-3p UMI counts and reads for individual patients (GBM [yellow] and GII-III [green]) are plotted (black dot); upper error bars signify the 90th percentile and lower error bars represent the 10th percentile, the middle line represents the median and the open square signifies the mean. (**C**) miR-486-3p expression levels in serum-EVs captured from GBM, GII-III and healthy controls (HC). (**D**) Word cloud for hsa-miR-486 (accession: MI0002470) generated by miRBase from 182 open access papers and 1080 associated sentences.

**Table 1 ijms-21-04954-t001:** Significant piRNAs with increased expression in GBM CUSA-EVs are previously associated with cancer.

piRNA ID	FC	Adjust. *p*-Value	Links to Cancer in the Literature
**hsa_piR_020829**	23.71	5.18 × 10^−2^	Significant increase in breast cancer versus normal tissue [55]
**hsa_piR_016659**	14.79	1.02 × 10^−2^	Significant increase in metastatic renal cell carcinoma tissue [56]
**hsa_piR_016658**	10.57	2.59 × 10^−2^	Increase distinguishes endometrial carcinogenesis from normal tissue [57]
**hsa_piR_020365**	8.52	4.32 × 10^−2^	Significant decrease in serum of colorectal cancer patients relative to healthy controls [58]
**hsa_piR_016677**	6.87	5.39 × 10^−2^	Significant decrease in endometrioid ovarian cancer versus normal tissue [59]
**hsa_piR_004153**	5.33	4.59 × 10^−2^	Significant association with recurrence free survival in breast cancer [55]
**hsa_piR_004987**	4.16	4.98 × 10^−2^	Significant increase in breast cancer tissue [60]
**hsa_piR_020485**	3.59	4.48 × 10^−2^	Significant increase in breast cancer [60]
**hsa_piR_020490**	2.44	5.43 × 10^−2^	Increase in GBM versus normal brain [61]
**hsa_piR_020541**	2.70	5.47 × 10^−2^	Significant association with overall survival and recurrence free survival in breast cancer [55]

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
