# Peer review of "Deep Sequencing of Small RNAs from Neurosurgical Extracellular Vesicles Substantiates miR-486-3p as a Circulating Biomarker that Distinguishes Glioblastoma from Lower-Grade Astrocytoma Patients"

_ijms, 2020, doi:10.3390/ijms21144954_

Round 1

Reviewer 1 Report

The ideas and the hypotheses are well addressed in this study and the obtained results are quite interesting and could help other researchers in the fields. I believe that the authors were successful in communicating their scientific message.

Regards

Author Response

Thank you for your review

Reviewer 2 Report

In this manuscript the authors evaluate the neurosurgical extracellular vesicle (EVs) content by deep sequencing of small RNAs in order to identify circulating biomarkers that discriminates glioblastoma from lower grade glioma. The issue of biomarkers charged in circulating, as important publications have proved, is of great interest and promising for assessing the molecular state of tumors. However, there are few points that should be addressed:

  1. The introduction is too long and dispersive and more suitable for a review; 90 references are too many. It should be more focused on the purpose of the experimental plan.
  2. Since the title of the manuscript focuses on miR-486-3p, the authors should discuss more in depth the potential role of miR-486-3p in glioblastoma. To this end, they could investigate miR-486-3p expression in their cohort of GBM and GII-III patients, if available. It would be also interesting to relate the findings of this manuscript to the previous study on miR-486-3p and GBM by Wu et al (PMID: 32086739). In Conclusion section, the authors should highlight the importance of the result obtained on miR-486-3p.
  3. Figure 1C: it would be better to visualize by TEM enriched EVs from GII-III CUSA along with enriched EVs from GBM CUSA.
  4. Figure 1D: this figure is not mentioned in the text.
  5. Table S1B: the authors showed the list of the Top-100 ExoCarta EV proteins. However, the exosome database ExoCarta provides the list of the top 100 proteins that are often identified in exosomes (http://exocarta.org/exosome_markers_new) rather than in extracellular vesicles (EVs). The authors should clarify this point. Nevertheless, since the authors characterized extracellular vesiscles, it may be more appropriate to show the list of the top 100 proteins that are often identified in EVs provided by Vesiclepedia (http://microvesicles.org/extracellular_vesicle_markers), and compare the 809 proteins identified as common to all patients with the list provided by Vesiclepedia.
  6. Results, 2.5 paragraph: the authors investigated sncRNAs identified in serum-EVs captured from a different cohort of patients and healthy control. However, they did not provide any information and details on the methods relative to this population of EVs, as well as on their characterization (e.g. how serum-EVs were isolated, size distributions and concentrations of serum-EV population). Please add these details or provide reference describing the method used for isolation of serum-EVs.
  7. Results, 2.5 paragraph: at the end of the paragraph, the authors stated “There was association between the magnitude of piR_016658 and piR_016659 changes and glioma severity”. However, they did not show any data supporting this conclusion. Please clarify this statement.
  8. Some errors need to be proofread:
  9. Results, line 173: 200 nm instead of 200 nt.
  10. Results, line 224: 11 piRNAs instead of 10 piRNAs, as the number of piRNAs listed in Table 1.

Author Response

1. We have shortened the Introduction and removed nearly half of the references. We hope this has improved the readability of this section.

2. We have added a new Figure 4 with boxplots for miR-486-3p expression for CUSA-EV samples (Figure 4A, UMI counts and Figure 4B, reads) and serum-EV samples (Figure 4C). A word cloud based on 182 open access papers and 1080 associated sentences that mention miR-486-3p has also been added to visualise prominent experimental disease and functional associations of miR-486-3p (Figure 4D). The following has been added to the text:

Analogous expression of miR-486-3p in CUSA (Figure 4A and 4B) and serum-EVs (Figure 4C) was observed, with higher levels in GBM, relative to GII-III. An miRBase word cloud [62] was generated for hsa-miR-486 and lists functional information extracted from 182 open access papers and 1080 associated sentences (Figure 4D). Prominent keywords include ‘cancer’, ‘increased’ and ‘plasma’ as well as ‘staging’, ‘survival’ and ‘prognosis’ (Figure 4D).” (lines 223-6)

We have proposed a potential link between miR-486-3p expression and IDH-mutational status and have amended the discussion to include associations to previous studies including Wu et al (PMID:32086739):

"Significantly high miR-486-3p levels were observed in both CUSA and serum-EVs from GBM patients (Figure 4). In our previous multivariate analysis, miR-486-3p was identified as one of the top four features that distinguish GBM from GII-III patient sera [18]. miR-486-3p is overexpressed in primary glioma tissues and its expression is strongly correlated with glioma grade and poor overall survival [87]. Moreover, increased miR-486-3p levels were linked to NF-κB activation which promotes glioma aggression [87]. The miR-486 gene is located in the intron of the sAnk1 gene [88]; the Ank1 promoter region contains TGF-β-responsive element (SRE) and tandem repeats of the TCF/LEF transcriptional responsive element, connecting miR-486 regulation with TGF-β/Smad and Wnt/β-catenin signalling [87]. IDH1-mutations, present in the GII-III cohort here, were shown to negatively regulate Wnt/β-catenin signalling [89], and TGF-β expression is lower in IDH-mutant compared to IDH-wildtype gliomas [90]. Thus, in addition to its potential use as a marker for glioma aggression, miR-486 levels may correspond to IDH-mutational status. miR-486-3p is also a marker for other aggressive tumour types, with high expression observed in oral squamous cell carcinoma tissue [91], in the CSF of medulloblastoma patients [92] and in colorectal cancer [93]. Paradoxically, other studies have reported that increased miR-486-3p expression promotes chemo-sensitivity of GBM to TMZ by targeting MGMT [94] and esophageal carcinoma cell-lines to 5-fluorouracil [95], suggesting important roles for miR-486-3p in targeting cell survival molecular pathways in response to cytotoxic stimuli. The aberrant expression of miR-486-3p across multiple pathologies and in response to chemotherapy makes it an attractive biomarker for monitoring of glioma progression and treatment responses [95]. Further study using larger and more homogenous glioma cohorts including specimens taken during recurrent progression will confirm whether miR-486-3p is a useful marker for glioma severity and progression." (lines 349-70)

3. Figure 1C now includes TEM images of CUSA-EVs enriched from GBM and astrocytoma grade II surgeries. The Figure legend has been amended:

“(c-1) Transmission electron microscopy (TEM) was used to visualize enriched CUSA-EVs from GBM and (c-2) astrocytoma grade II surgeries, scale bar=200 nm.” (lines 159-61)

In accordance with MISEV2018 guidelines, which recommends that a ratio of two different EV quantitation methods are presented, we have also provided the particle to protein ratio for the twelve EV fractions in Figure 1A.

4. We apologise for this oversight, Figure 1D is now mentioned line 131.

5. Supplementary table S1B has been edited and now lists the top 100 EV proteins listed in Vesiclepedia. Exocarta has been replaced in the manuscript text with Vesiclepedia (line 131)

6. We have added the following: “The serum-EV cohorts used in this study are detailed in Table S2A. The methods for serum processing, EV isolation, EV characterisation, sncRNA preparation and sequencing are previously described [18]." (line 537-9).

7. We have clarified this statement:

“While there were no significant distinguishing piRNA species identified between GBM and GII-III serum-EVs, a trend was observed between the magnitude of piR_016658 and piR_016659 changes and glioma severity (Figure 5C-1 and C-2); where HCs showed the highest expressions of piR_016658 (Figure 5C-2) and piR_016659 (Figure 5C-1), reduced levels were measured in GII-III patients and greater reductions observed in the GBM cohort.” (lines 270-4)

8. and 9. We apologise for these errors, nm has been corrected to nt and 11 piRNAs corrected to 10 piRNAs.

Round 2

Reviewer 2 Report

The authors addressed all of my concerns.